# A Novel Model-Based Attribute Inference Attack in Federated Learning

**Ilias Driouich**
ilias.driouich@inria.fr
ilias.driouich@amadeus.com
Inria, Univ. Côte d'Azur
Amadeus

**Chuan Xu**
chuan.xu@inria.fr
Inria, Univ. Côte d'Azur

**Giovanni Neglia**
giovanni.neglia@inria.fr
Inria, Univ. Côte d'Azur

**Frederic Giroire**
frederic.giroire@cnrs.fr
Inria, Univ. Côte d'Azur, CNRS, I3S

**Eoin Thomas**
eoin.thomas@amadeus.com
Amadeus

## Abstract

In federated learning, clients such as mobile devices or data silos (e.g. hospitals and banks) collaboratively improve a shared model, while maintaining their data locally. Multiple recent works show that client's private information can still be disclosed to an adversary who just eavesdrops the messages exchanged between the targeted client and the server. In this paper, we propose a novel model-based attribute inference attack in federated learning which overcomes the limits of gradient-based ones. Furthermore, we provide an analytical lower-bound for the success of this attack. Empirical results using real world datasets confirm that our attribute inference attack works well for both regression and classification tasks. Moreover, we benchmark our novel attribute inference attack against the state-of-the-art attacks in federated learning. Our attack results in higher reconstruction accuracy especially when the clients' datasets are heterogeneous (as it is common in federated learning). Most importantly, our model-based fashion of designing powerful and explainable attacks enables an effective quantification of the privacy risk in FL.

## 1 Introduction

Federated learning (FL) enables multiple clients to collaboratively train a better global model [1, 2, 3]. As clients' data is not collected by a third party, FL naturally offers a certain level of privacy. Nevertheless, recent works have demonstrated that FL may not provide formal privacy guarantees. Especially, an (honest-but-curious) adversary can infer some sensitive client information just by eavesdropping the exchanged messages. In fact, when the adversary has access to the clients' model **gradients** in FL,[1] it can reconstruct private training samples (e.g., images) as shown by [4, 5, 6, 7, 8]. This sample-reconstruction attack works well when gradients are calculated on extremely small batches or when the data points belonging to the same class are similar, e.g., personal images of the same person or images of the same digit in MNIST dataset. Moreover, the leaked gradients in FL could trigger an attribute inference attack (AIA). In [4] the authors propose an AIA which picks the sensitive attributes that minimize the Euclidean distance between the virtual gradients and the true gradients by L-BFGS algorithm [9]. In recent work [10] the authors propose an AIA that aims to

---

[1]As the adversary (which could be the server) is assumed to know the learning rate, it can then decode the local update of the victim, i.e., the gradient (when client's local step is 1) or the sum of the gradients (when client's local step is larger than 1).

36th Conference on Neural Information Processing Systems (NeurIPS 2022).

minimize the cosine similary and assumes that the sensitive attributes are discrete random variables. The resulting optimization problem is solved by the reparametrization trick - Gumbel softmax [11]. As both these attacks target per-sample attribute values, while model updates merge the information of multiple samples, the attacks perform poorly on large datasets. In particular, the attack accuracy drops by almost half on the Genome dataset [12] when the client's local dataset size increases from 50 to 1000 samples [10, Table 9].

All the above-mentioned **gradient-based** attacks for FL rely on reconstructed gradients, and thus their accuracies are sensitive to the batch size and the local dataset size. Alternatively, the adversary could use directly the models exchanged between the server and the clients to perform some attacks in a **model-based** way, such as the model inversion attack [13, 14] or the attribute inference attack [15, 16, 17] originally studied in the centralized setting. However, these model-based attacks work well only when the model over-fits the client's local data, which is not the case for the models exchanged between the FL clients and the server. In fact, in FL, clients cannot perform many local gradient steps between two consecutive communication rounds without the risk of harming convergence of the global model due to the heterogeneity of clients' local datasets [1, 18]. Consequently, the returned models from the clients are far from their local optima. Besides, the optimal global model and the optimal local one may be far apart [19, Figures 2,3]. As such, there is little value in attacks on the global model. In this regard the authors in [20] have initiated the study of a new attack, called the local model reconstruction attack, where the adversary seeks to reconstruct the model, a client would have trained using only its local dataset. This attack is potentially very dangerous as its performance doesn't degrade when the batch size or the local number of steps increases [20, Table 1], allowing the adversary to trigger the above-mentioned model-based attacks with better performance.

In this paper, we propose a novel model-based attribute inference attack leveraging the local model reconstruction attack, which overcomes the limits of gradient-based attribute inference attacks, i.e, its performance does not decrease when the number of local steps, and the local dataset size increase. We assume a weak adversary, who is honest-but-curious, i.e, who only eavesdrops the exchanged messages between the client and the server, but does not interfere with the training process. The adversary (e.g., a malicious server) knows the structure of the trained model and the loss function, as well as the training algorithm, as commonly assumed in the literature [21, 6, 7, 8, 22].

Our main contributions can be summarized as follows:

- We prove that there exists a dataset with a binary sensitive attribute on which no gradient-based attribute inference attack in FL can achieve more than $50\%$ accuracy. Furthermore, we propose a novel model-based attribute inference attack in FL and we provide an analytical lower bound for its accuracy, when training a linear least squares problem with full batch size and one local step (Sec. 4).
- We benchmark our model-based attribute inference attack against the state-of-the-art gradient-based ones for FL and measure an improvement of 6 to 10 percentage points in the attack accuracy (Sec. 5).

## 2 Background

We denote by $\mathcal{C}$ the set of all clients participating to FL. Let $\mathcal{D}_c$ be the local dataset of client $c \in \mathcal{C}$ drawn from a universe $\mathcal{Z}$ and $|\mathcal{D}_c|$ denote the size of $\mathcal{D}_c$. In FL, clients cooperate to learn a global model, which minimizes the following empirical risk over all the data owned by clients:

$$\min_{\theta \in \mathbb{R}^d} \mathcal{L}(\theta) = \sum_{c \in \mathcal{C}} p_c \mathcal{L}_c(\theta) = \sum_{c \in \mathcal{C}} p_c \left( \frac{1}{|\mathcal{D}_c|} \sum_{x \in \mathcal{D}_c} l(\theta, x) \right), \tag{1}$$

where $l(\theta, x) : \mathbb{R}^d, \mathcal{Z} \to \mathbb{R}_+$ measures the loss of the model $\theta$ on the sample $x \in \mathcal{Z}$ and $p_c$ is the positive weight of client $c$, s.t. $\sum_{c \in \mathcal{C}} p_c = 1$. Common choices of weights are $p_c = \frac{1}{|\mathcal{C}|}$ or $p_c = \frac{|\mathcal{D}_c|}{\sum_{c \in \mathcal{C}} |\mathcal{D}_c|}$.

Let $\theta^* = \arg\min_{\theta \in \mathbb{R}^d} \mathcal{L}(\theta)$ be a global optimal model, i.e., a minimizer of Problem (1). A general framework to learn such a global model in a federated way is shown in Algo. 1; it generalizes a large number of FL algorithms, including FedAvg [1], FedProx [3], and FL with different client sampling techniques [23, 24, 25]. The model $\tilde{\theta} = \theta(T)$—the output of Algo. 1—is the tentative solution of

problem (1). Its performance depends on the specific FL algorithm, which precises how clients are selected in line 2, how the updated local models are aggregated in line 5, and how the local update rule works in line 8. For example, in FedAvg [1], clients are selected uniformly at random from the available clients, the local models are averaged with constant weights, and the clients perform locally multiple stochastic gradient steps as precised in Algo. 2.

---

**Algorithm 1** Framework for cross-device federated learning

---

**Output**: $\theta(T)$

Server: {global model $\theta \in \mathbb{R}^d$, local models $\{\theta_c \in \mathbb{R}^d,\ c \in \mathcal{C}\}$.}
 1: **for** $t \in \{0, ..., T-1\}$ **do**
 2:     Server selects a subset of the clients $\mathcal{C}_s(t) \subseteq \mathcal{C}$,
 3:     Server broadcasts the current global model $\theta(t)$ to $\mathcal{C}_s(t)$,
 4:     Server waits for the updated local models $\theta_c$ from every client $c \in \mathcal{C}_s(t)$,
 5:     Server updates $\theta(t+1)$ by aggregating the received updated local models.
Client $c \in \mathcal{C}$: {global model $\theta$, local model $\theta_c$, local dataset $\mathcal{D}_c$}
 6: **while** FL training is not completed **do**
 7:     Client listens for the arrival of new global model $\theta$,
 8:     Client updates its local model: $\theta_c \leftarrow \text{Local\_Update}^c(\theta, \mathcal{D}_c)$
 9:     Client sends back $\theta_c$ to the server.

---

---

**Algorithm 2** Client $c$'s local update rule in FedAvg [1]

---

$\text{Local\_Update}^c(\theta, \mathcal{D}_c)$
$\theta$: server model, $\mathcal{D}_c$: local dataset, $B$: batch size, $E$: the number of local epochs, $\eta$: learning rate.
 1: $\theta_c \leftarrow \theta, \mathcal{B} \leftarrow$ (split $\mathcal{D}_c$ into batches of size $B$)
 2: **for** each local epoch $e$ from 1 to $E$ **do**
 3:     **for** batch $b \in \mathcal{B}$ **do**
 4:         $\theta_c \leftarrow \theta_c - \eta \times \mathbf{g}(\theta_c, b), \quad$ where $\mathbf{g}(\theta_c, b) = \frac{1}{B} \sum_{x \in b} \nabla l(\theta_c, x)$
 5: Return $\theta_c$

---

## 3  Gradient-based AIA

AIA aims to infer a sensitive attribute (e.g., health status, political preference or income), given access to some *public* attributes (e.g., age and gender). It was first studied under the cloud environment in which the adversary is assumed to have full access to the released ML model (white-box attack) or only to its predictions (black-box attack) [13, 14, 15, 16]. The basic idea for the attack is to invert the ML model to get the most likely value for the sensitive attribute, given the information available. For the FL scenario in which the adversary has solely access to the exchanged models, only one work, [10], proposes to perform a gradient-based AIA. Basically, their idea is to find the sensitive attributes for which the resulting virtual gradient is the closest to the true gradient in terms of cosine similarity.

In this paper, we consider a simple AIA scenario (the same as in [10]) where there is only one sensitive attribute in the data. Let $\mathcal{D}_c = (\mathbf{X}_c^p, \mathbf{s}, \mathbf{y}_c)$ be the local dataset of client $c$, where $\mathbf{X}_c^p \in \mathbb{R}^{m \times (d-1)}$ is the matrix of the public attributes, $\mathbf{s} \in \mathbb{R}^m$ is the vector of the sensitive attribute values and $\mathbf{y}_c \in \mathbb{R}^m$ is the vector of target values.

The adversary has access to $\mathbf{X}_c^p$ and $\mathbf{y}_c$, and aims to reconstruct $\mathbf{s}$ by eavesdropping the FL training process.[2]

Additionally, in common gradient-based attribute inference attacks proposed for FL settings [4, 10] in which gradients are computed on large data, authors have shown that it becomes harder to reconstruct per-example attributes when the victim participant has more data. On the other hand, the choice of gradients is decisive for the attack's success. In the following we prove that there exists an extreme case where all the gradient-based attacks can not achieve more than 50% success rate.

---

[2]Different from [13] and [16], we consider a rather weak adversary as in [10] who has neither the approximation of the error distribution, nor the distribution over the data points.

**Proposition 1.** *For any AIA for FL, there exists a dataset on which the attack can reconstruct a binary sensitive attribute with at most $50\%$ accuracy.*

*Proof.* Appendix A.1. $\square$

## 4 Our Model-Based AIA for FL

In this section, we first propose a new AIA for linear least squares regression in FedAvg when the sensitive attribute is binary (Algo. 3). We show in Proposition 3 that its accuracy can be lower bounded through the parameters of the reconstructed local model. Finally, we propose an AIA (Algo. 4) without analytical performance guarantees but suited for any domain of the sensitive attribute and for any machine learning problem.

For the linear least squares regression in FedAvg, the authors have shown in [20, Observation 1] that the adversary can decode the exact local optimal model (Line 1 of Algo. 3). A natural approach for AIA on a binary sensitive attribute is to choose for every sample the sensitive value which minimizes the least square error of the local model (Line 3 of Algo. 3) and, then, round the value to the closest number in $\{0, 1\}$. Alternatively, when considering FedAvg with one local step, the adversary can obtain the prior distribution of the sensitive attribute from the gradients (Line 2 of Algo. 3), i.e., the percentage of ones in the true sensitive vector $\mathbf{s}$. It can then use this information to improve the rounding scheme, by guaranteeing that the estimation vector $\hat{\mathbf{s}}$ has the same number of ones as in $\mathbf{s}$ (Line 4 of Algo. 3).

---

**Algorithm 3** Attribute inference attack on binary attribute for linear least squares regression in FedAvg

---

**Input**: $\mathcal{M}^c$: the messages exchanged between client $c$ and the server, $\mathbf{X}_c^p \in \mathbb{R}^{m \times (d-1)}$: the public attributes of client $c$, $\mathbf{y}_c \in \mathbb{R}^m$: public labels.

1: Decode the local model $\theta_c^* \in \mathbb{R}^d$ ([20, Observation 1])
2: Decode the sensitive attribute's distribution $\rho$, i.e., the percentage of ones in the local dataset (Proposition 2)
3: Let $\theta_c^{*,s} \in \mathbb{R}$ be the model parameter corresponding to the sensitive attribute and $\theta_c^{*,p} \in \mathbb{R}^{d-1}$ be the parameters corresponding to the public attributes, solve the following optimization problem:

$$\tilde{\mathbf{s}} = \arg \min_{\mathbf{s} \in \mathbb{R}^m} ||\mathbf{X}_c^p \theta_c^{*,p} + \mathbf{s}\theta_c^{*,s} - \mathbf{y}_c||_2^2$$

4: For every instance $i \in \{1, ..., m\}$:

$$\hat{s}_i = \begin{cases} 1 & \text{if } \tilde{\mathbf{s}}[i] \text{ belongs to the highest} \lceil \rho \times m \rceil \text{ values in } \tilde{\mathbf{s}} \\ 0 & \text{otherwise} \end{cases} \quad (2)$$

5: Return $\hat{\mathbf{s}}$ as the estimated sensitive attributes.

---

**Proposition 2.** *Consider training a least squares linear regression through FedAvg with full batch size and one local step and assume that a client's design matrix has its rank equal to the number of features. Once the client has communicated with the server $d + 1$ times, the adversary can infer the victim's sensitive attribute prior distribution in $O(d^4)$ operations.*

*Proof.* Appendix A.1 $\square$

**Proposition 3.** *Consider training a least squares linear regression through FedAvg with full batch size and one local step and assume that a client's design matrix has its rank equal to the number of features. Let $MSE$ be the mean square error of the local model $\theta_c^*$ and $\theta_c^{*,s}$ be the local model parameter corresponding to the sensitive attribute. Let $\rho$ be the (unknown) percentage of ones in the sensitive attributes $\mathbf{s}$. The adversary can infer the victim's sensitive attribute $\mathbf{s}$ in $O(d^4)$ operations with an accuracy larger than or equal to $\max\{|1 - 2\rho|, 1 - \frac{4MSE}{(\theta_c^{*,s})^2}\}$.*

*Proof.* Appendix A.1 $\square$

From Proposition 3, we see that if the local model fits well the known local data or the sensitive attribute has significant importance, the attack can achieve almost $100\%$ of the accuracy on these data. The discussion of how this model-based attack performs on the dataset that no graident-based method can achieve more than $50\%$, is moved to App. A.1.

We now propose a heuristic for AIA, Algo. 4, adapted to any sensitive attribute's domain, FL algorithm, and ML problem. Here, we just adopt the intuitive approach of minimizing the least square error on the decoded model for every data sample.

---

**Algorithm 4** Attribute inference attack in FL for general case

---

**Input**: $\mathcal{M}^c$: the messages exchanged between client $c$ and the server, $\mathbf{X}_c^p \in \mathbb{R}^{m \times (d-1)}$: the public attributes of client $c$, $\mathbf{y}_c \in \mathbb{R}^m$: corresponding public labels, $\mathbb{D}$: the domain of the sensitive attribute.

1: Decode the local model $\theta_c^* \in \mathbb{R}^d$ ( [20, Algo. 3])
2: Let $\mathbf{F}$ be the local model function, solve the following optimization problem:

$$\hat{\mathbf{s}} = \arg \min_{\mathbf{s} \in \mathbb{D}^m} ||\mathbf{F}(\theta_c^*, \mathbf{X}_c^p, \mathbf{s}) - \mathbf{y}_c||_2^2.$$

3: Return $\hat{\mathbf{s}}$ as the estimated sensitive attributes.

---

## 5 Experiments

**Experimental setup** We implemented all algorithms in PyTorch. Our experiments follow the standard structure of FedAvg [1]. Hyperparameters used in all experiments can be found in Appendix A.2.

**Flight Prices Dataset** This dataset contains booking details like origin, destination, booking time before departure, departure date, journey type (round trip or not), airline codes, and segment price, which corresponds to the target in the considered regression task. Similar datasets [26] have been used to model traveller preferences and their price elasticity. The data is split according to airlines resulting in 10 clients.

**Adult [27]** This dataset contains individual information such as sex, age, education level, family situation, working class, etc. This information is used to predict whether a person has an income higher than 50k\$. Details on how data is split across the 10 clients are given in Appendix A.2.

| | client ID | 0 | 1 | 2 | 3 | 4 | 5 | 6 | 7 | 8 | 9 | $\frac{\sum Acc_c}{10}$ |
|---|---|---|---|---|---|---|---|---|---|---|---|---|
| **Flight Prices** | LBFGS | 58.7 | 60.3 | 65.6 | 69.4 | 68.1 | 66.3 | 68.5 | 77.3 | **71.2** | 68.3 | 67.3 |
| | Gumbel | 60.2 | 62.6 | 66.8 | 70.1 | 58.3 | 72.7 | 63.4 | 69.2 | 71.0 | 74.9 | 66.9 |
| | Ours$_{\text{Global}}$ | 59.5 | 82.2 | 70.3 | 71.4 | 69.7 | 66.7 | **72.6** | 77.3 | 68.6 | 82.6 | 73.5 |
| | Ours$_{\text{LastReturned}}$ | 81.2 | 73.8 | 71.4 | 70.2 | 67.6 | 69.3 | 71.2 | 73.0 | 65.2 | 62.9 | 70.6 |
| | Ours$_{\text{Local}}$ | **97.7** | **84.4** | **77.3** | **75.9** | **80.2** | **79.9** | 68.9 | **78.3** | 59.6 | **89.9** | **76.9** |
| | client ID | 0 | 1 | 2 | 3 | 4 | 5 | 6 | 7 | 8 | 9 | $\frac{\sum Acc_c}{10}$ |
| **Adult** | LBFGS | 75.2 | 76.5 | 60.3 | 61.2 | 60.4 | 60.9 | 61.2 | 60.6 | 60.0 | 62.3 | 67.2 |
| | Gumbel | 62.2 | 67.7 | 61.1 | 68.0 | 68.2 | 57.3 | 64.3 | 64.3 | 64.3 | 64.3 | 64.2 |
| | Ours$_{\text{Global}}$ | 69.9 | 77.5 | 80.2 | 61.3 | 61.3 | 62.3 | 60.0 | 62.1 | 60.3 | 60.6 | 65.6 |
| | Ours$_{\text{LastReturned}}$ | 71.2 | 72.3 | 73.8 | 60.4 | 65.6 | 69.0 | 61.5 | 62.0 | 60.1 | 62.3 | 65.8 |
| | Ours$_{\text{Local}}$ | **79.4** | **78.4** | **83.5** | **69.9** | **69.4** | **69.2** | **70.3** | **70.5** | **70.6** | **70.9** | **73.7** |

Table 1: AIA reconstruction accuracy per client under different methods. We consider two gradient-based attacks: LBFGS and Gumbel, and three model-based attacks: Ours$_{\text{Global}}$, Ours$_{\text{LastReturned}}$, Ours$_{\text{Local}}$ which correspond to our AIA triggered by the final global model, the last returned model and the decoded local model, respectively.

In our experiments, we considered as sensitive attribute the journey type for the Flight Prices dataset and the gender for the Adult dataset. We compare our attack with two gradient-based AIA attacks: LBFGS [4] and the more recent work (called Gumbel in our paper) [10].

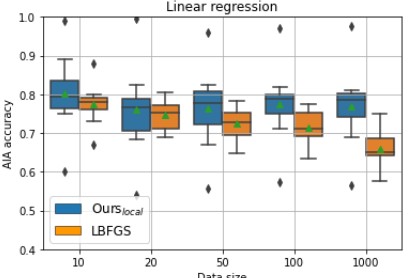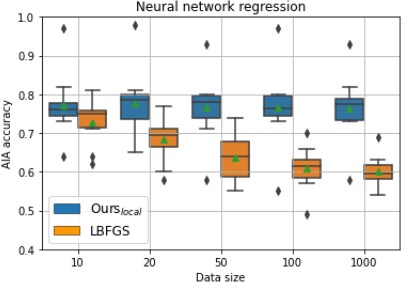

Figure 1: Flight Prices: Influence of the local dataset size on the AIA accuracy.

In addition, we conduct two other model-based attacks which follow the same procedure as in Algo. 3 and Algo. 4, but replace the local model with the final global model and the last returned model, respectively. Note that, for LBFGS and Gumbel, the choices of the considered gradients are critical to the attack's performance [10]. We tune the set of the gradients considered for every client and show the choices in Tables 6 and 7 of App. A.2.

In Table 1, we show that our attack triggered by the decoded local model enjoys a higher average reconstruction accuracy than gradient-based attacks (on average 10 percentage points improvement for Flight Prices when training a linear least square regression and 6.5 percentage points for Adult). Besides, experiment on Flight Prices dataset show that, our lower bound (Prop. 3) is a very good indicator on the attack accuracy for each client (See Fig. 3 in App. A.2).

Next, we study the performance of AIA under different local dataset sizes. To build such a scenario, we randomly select a fixed number of samples from each client's local dataset. We can see from Figure 1 that the advantage of our AIA triggered by the decoded local model compared with LBFGS increases, when the dataset size increases for both the linear model and neural network. Besides, our attack is insensitive to the local data size. While the average accuracy for LBFGS (as indicated by the green triangles) drops from $0.78$ to $0.67$ for the linear model and from $0.74$ to $0.6$ for the neural network, when the dataset size increases from 10 to 1000. Gradient-based AIA performance is worsened with larger datasets as the gradients (local updates) in FL merge the information of data samples.

## 6   Conclusion

In this paper, we propose a novel model-based attribute inference attack for FL. This attack does not suffer due to larger dataset sizes, unlike gradient-based attacks previously proposed in the literature do. Most crucially, our model-based attack provides a new angle for designing powerful and explainable attacks to effectively quantify the privacy risk in FL. Our next research step will be to evaluate the performance of other classic attacks (e.g., membership inference attack and model inversion attack) in FL when carried out on the reconstructed local model.

In addition, all the passive attacks designed for FL till now (including ours) do not work under a secure aggregation protocol which allows the server—at the a price of additional computation and communication—to aggregate the local updates without having access to each individual update. Further research is required to evaluate potential privacy leakages also in this setting.

## Acknowledgements

This work has been supported by the French government, through the France relance plan managed by the National Research Agency (ANR) with the reference number ANR-21-PRRD-0005-01. This work was also supported by Amadeus and the Inria Sophia Antipolis - Méditerranée, "NEF" computation cluster .

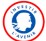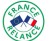

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

# A Appendix

## A.1 PROOFS

**Proof of Proposition 1**

*Proof.* Here, we consider training a least squares linear regression through FedAvg with full batch size and with one local step. Remember that a design matrix contains as rows the input features of the samples in training dataset $\mathcal{D}_c$. Let $\mathbf{X}_c \in \mathbb{R}^{m \times d}$ be the design matrix of client $c$ and $\mathbf{y}_c \in \mathbb{R}^m$ be the labels of the local dataset $\mathcal{D}_c$ with size $m = |\mathcal{D}_c|$.

$$\mathcal{L}_c(\theta) = \frac{\|\mathbf{X}_c\theta - \mathbf{y}_c\|^2}{m} \tag{3}$$

We know that $\theta_c^* = (\mathbf{X}_c^T\mathbf{X}_c)^{-1}\mathbf{X}_c^T\mathbf{y}_c$. When the batch size is set to $m$ in FedAvg, the gradient is given by:

$$\mathbf{g}(\theta) = \frac{2}{m}\left(\mathbf{X}_c^T\mathbf{X}_c\theta - \mathbf{X}_c^T\mathbf{y}_c\right). \tag{4}$$

$\mathbf{X}_c$ can be decomposed as $[\mathbf{X}_c^p \mathbf{s}]$ where $\mathbf{X}_c^p \in \mathbb{R}^{m \times (d-1)}$ is the matrix of public attributes and $\mathbf{s} \in \{0, 1\}^m$ is the vector of the sensitive binary attributes. Replacing $\mathbf{X}_c = [\mathbf{X}_c^p \mathbf{s}]$ in (4), the gradient that the adversary obtained by eavesdropping the victims' exchanged models is:

$$\mathbf{g}(\theta) = \frac{2}{m}\left(\begin{bmatrix}(\mathbf{X}_c^p)^T\mathbf{X}_c^p & (\mathbf{X}_c^p)^T\mathbf{s} \\ \mathbf{s}^T\mathbf{X}_c^p & \mathbf{s}^T\mathbf{s}\end{bmatrix}\theta - \begin{bmatrix}(\mathbf{X}_c^p)^T\mathbf{y}_c \\ \mathbf{s}^T\mathbf{y}_c\end{bmatrix}\right). \tag{5}$$

Now, to show the impossibility result, we will demonstrate two local datasets whose binary sensitive attributes are all different but whose gradients, according to (5), are exactly the same.

**Dataset A**: The design matrix on the public attributes $\mathbf{X}_c^p(A) \in \mathbb{R}^{m \times (d-1)}$ can be decomposed into two equivalent parts, s.t., $\mathbf{X}_c^p(A) = \begin{bmatrix}\mathbf{Z}_c \\ \mathbf{Z}_c\end{bmatrix}$ and $\mathbf{Z}_c \in \mathbb{R}^{\frac{m}{2} \times (d-1)}$. The vector of sensitive attributes $\mathbf{s}^A$ is decomposed as $\mathbf{s}^A = \begin{bmatrix}\mathbf{1} \\ \mathbf{0}\end{bmatrix}$, where $\mathbf{1}$ is the vector of all ones of size $\frac{m}{2}$ and $\mathbf{0}$ is the vector of all zeros of size $\frac{m}{2}$. The corresponding decomposed labels $\mathbf{y}_c(A) = \begin{bmatrix}\mathbf{y}_c^1 \\ \mathbf{y}_c^2\end{bmatrix}$ satisfies $\mathbf{1}^T\mathbf{y}_c^1 = \mathbf{1}^T\mathbf{y}_c^2$.

**Dataset B**: The design matrix on the public attributes $\mathbf{X}_c^p(B) = \mathbf{X}_c^p(A)$. The labels $\mathbf{y}_c(B) = \mathbf{y}_c(A)$. The sensitive attributes are opposite to the ones in Dataset A, i.e., $\mathbf{s}^B = \begin{bmatrix}\mathbf{0} \\ \mathbf{1}\end{bmatrix}$.

First, since the public design matrix $\mathbf{X}_c^p$ and the labels $\mathbf{y}_c$ are the same for two dataset, the part $(\mathbf{X}_c^p)^T\mathbf{X}_c^p$ and $(\mathbf{X}_c^p)^T\mathbf{y}_c$ in (5) are the same. Besides $(\mathbf{s}^A)^T\mathbf{s}^A = (\mathbf{s}^B)^T\mathbf{s}^B = \frac{m}{2}$ and $(\mathbf{X}_c^p(A))^T\mathbf{s}^A = (\mathbf{X}_c^p(B))^T\mathbf{s}^B = \mathbf{Z}_c^T\mathbf{1}$. At last, since $\mathbf{y}_c(A) = \mathbf{y}_c(B) = \begin{bmatrix}\mathbf{y}_c^1 \\ \mathbf{y}_c^2\end{bmatrix}$ and $\mathbf{1}^T\mathbf{y}_c^1 = \mathbf{1}^T\mathbf{y}_c^2$, we have $(\mathbf{s}^A)^T\mathbf{y}_c(A) = (\mathbf{s}^B)^T\mathbf{y}_c(B) = \mathbf{1}^T\mathbf{y}_c^1$.

Thus, we conclude that the gradients for the dataset A and the dataset B are the same and by observing the gradients, the adversary could not distinguish these two dataset whose sensitive attributes are totally opposite. As a result, any gradient-based attack on this binary sensitive attribute can not achieve more than $50\%$ accuracy. $\square$

**Proof of Proposition 2**

*Proof.* Let $\mathbf{X}_c \in \mathbb{R}^{m \times d}$ be the design matrix with rank $d$ and $\mathbf{y}_c \in \mathbb{R}^m$ be the labels in the local dataset $\mathcal{D}_c$ of the victim $c$. Remark that in (5), $\mathbf{s}^T\mathbf{s}$ represents the number of ones in binary sensitive attribute $\mathbf{s}$. Therefore, once the adversary gets $d + 1$ exchanged messages, in the same fashion as in [20, Observation 1], it can reconstruct the exact matrix $\begin{bmatrix}(\mathbf{X}_c^p)^T\mathbf{X}_c^p & (\mathbf{X}_c^p)^T\mathbf{s} \\ \mathbf{s}^T\mathbf{X}_c^p & \mathbf{s}^T\mathbf{s}\end{bmatrix}$ and in particular the victim's sensitive attribute prior distribution. $\square$

**Proof of Proposition 3**

*Proof.* From [20, Observation 1] the adversary can recover the exact local model $\theta_c^*$ and from 2 the adversary can recover the victim's sensitive attribute prior distribution (thus the value of $\rho$). Let $\mathcal{P}$ be the set of the indices of the instances whose sensitive attributes are ones and $\mathcal{N}$ be the set of the indices of the instances whose sensitive attributes are zeros. Let $\mathbf{e}$ be the vector of residuals for the local data, i.e., $\mathbf{e} = \mathbf{y}_c - (\mathbf{X}_c^p \theta_c^{*,p} + \mathbf{s}\theta_c^{*,s})$ where $\mathbf{s}$ is the true labels of the sensitive attribute. Solving the optimization problem in line 3 Algo. 3, we get

$$\tilde{\mathbf{s}} = \arg\min_{\mathbf{s} \in \mathbb{R}^m} ||\mathbf{X}_c^p \theta_c^{*,p} + \mathbf{s}\theta_c^{*,s} - \mathbf{y}_c||_2^2 = \frac{\mathbf{y}_c - \mathbf{X}_c^p \theta_c^{*,p}}{\theta_c^{*,s}}.$$

Therefore, we have

$$\mathbf{s} = \frac{\mathbf{y}_c - \mathbf{X}_c^p \theta_c^{*,p} - \mathbf{e}}{\theta_c^{*,s}} = \tilde{\mathbf{s}} - \frac{\mathbf{e}}{\theta_c^{*,s}}. \tag{6}$$

For an instance $j \in \mathcal{N}$, we have $\tilde{\mathbf{s}}[j] = \frac{\mathbf{e}[j]}{\theta_c^{*,s}}$. For the instance $i \in \mathcal{P}$, $\tilde{\mathbf{s}}[i] = 1 + \frac{\mathbf{e}[i]}{\theta_c^{*,s}}$. The necessary condition for a false prediction on instance $j$ according to the rule (2) is that, there exists $i \in \mathcal{P}$ s.t. $\tilde{\mathbf{s}}[j] > \tilde{\mathbf{s}}[i]$, i.e., $\frac{\mathbf{e}[j]}{\theta_c^{*,s}} > 1 + \frac{\mathbf{e}[i]}{\theta_c^{*,s}}$. It follows that $\min\{\frac{|\mathbf{e}[j]|}{\theta_c^{*,s}}, \frac{|\mathbf{e}[i]|}{\theta_c^{*,s}}\} \geq \frac{1}{2}$.

Meanwhile, we have

$$\sum_{i \in \mathcal{P}} \frac{\mathbf{e}[i]^2}{(\theta_c^{*,s})^2} + \sum_{j \in \mathcal{N}} \frac{\mathbf{e}[j]^2}{(\theta_c^{*,s})^2} = \frac{MSE \times m}{(\theta_c^{*,s})^2}. \tag{7}$$

Then each mispredicted pair, contributes to (7) by at least $1/2$.

Hence the maximum number of pairs $(i, j)$ mispredicted could be no more than $\frac{2MSE \times m}{(\theta_c^{*,s})^2}$. Therefore, the success of the attack is larger than or equal to $1 - \frac{4MSE}{(\theta_c^{*,s})^2}$.

On the other hand, by the nature of rule (2), the attack accuracy is also lower bounded by $|1 - 2\rho|$.

□

**Discussion on the performance of our model-based attack on the dataset built for the impossibility result**

In App. A.1, we have described two datasets, such that any gradient-based attack cannot achieve more than 50% accuracy on both of them. More precisely, if the attack works well on one dataset, it definitely works poorly on the other one.

Here, we study how our model-based attack performs on such datasets. Actually, since the local optimal model is $\theta_c^* = (\mathbf{X}_c^T \mathbf{X}_c)^{-1} \mathbf{X}_c^T \mathbf{y}_c$, the two datasets considered in Prop. 1 lead to the same local optimal model. Therefore, our model-based attack cannot achieve more than 50% accuracy on both of them neither. More generally speaking, since the messages observed by the adversary during the training of FedAvg are exactly the same for these two datasets, no attribute inference attack to FedAvg training can achieve more than 50% accuracy on these two datasets.

However, our model-based attack overcomes other limits of gradient-based attack, e.g., its sensitivity to the local dataset size. Attribute inference attack is essentially a post-hoc per-example task. Our model-based attack performs on a per-example basis as, for each data sample, it chooses the value of the sensitive attribute which minimizes the least square error of the local model. Therefore, it is not affected by the dataset size (as shown experimentally in Fig. 1). On the contrary the gradients merge information from all the samples, the gradient-based attacks suffer from the large local dataset.

## A.2 ADDITIONAL EXPERIMENTS DETAILS AND RESULTS

In this section, we provide supplementary details on our experimental setup (Sec. A.2), additional datasets description (Sec. A.2) and additional explanations for the experiments (Sec. A.2).

**Experimental Setup**

For all our FL trainings, the learning rate is set to 10e-2 and the number of communication rounds is set to 1000. Tables 2 and 3 give the details on hyper-parameters used in our methods for attribute inference attack LBFGS [4] and Gumbel [10]

| | client ID | 0 | 1 | 2 | 3 | 4 | 5 | 6 | 7 | 8 | 9 |
|---|---|---|---|---|---|---|---|---|---|---|---|
| **LBFGS** | Epochs | 10e4 | 10e4 | 10e4 | 10e4 | 10e4 | 10e4 | 10e4 | 10e4 | 10e4 | 10e4 |
| | Learning rate | 10e-1 | 10e-1 | 10e-1 | 10e-1 | 10e-1 | 10e-1 | 10e-1 | 10e-1 | 10e-1 | 10e-1 |
| **Gumbel** | client ID | 0 | 1 | 2 | 3 | 4 | 5 | 6 | 7 | 8 | 9 |
| | Epochs | 10e4 | 10e4 | 10e4 | 10e4 | 10e4 | 10e4 | 10e4 | 10e4 | 10e4 | 10e4 |
| | Learning rate | 10e2 | 10e2 | 10e2 | 10e2 | 10e2 | 10e2 | 10e2 | 10e2 | 10e2 | 10e2 |

Table 2: Hyper-parameters used for the attacks on Flight Prices dataset.

| | client ID | 0 | 1 | 2 | 3 | 4 | 5 | 6 | 7 | 8 | 9 |
|---|---|---|---|---|---|---|---|---|---|---|---|
| **LBFGS** | Epochs | 10e4 | 10e4 | 10e4 | 10e4 | 10e4 | 10e4 | 10e4 | 10e4 | 10e4 | 10e4 |
| | Learning rate | 10e-2 | 10e-2 | 10e-2 | 10e-2 | 10e-1 | 10e-2 | 10e-2 | 10e-2 | 10e-2 | 10e-2 |
| **Gumbel** | client ID | 0 | 1 | 2 | 3 | 4 | 5 | 6 | 7 | 8 | 9 |
| | Epochs | 10e4 | 10e4 | 10e4 | 10e4 | 10e4 | 10e4 | 10e4 | 10e4 | 10e4 | 10e4 |
| | Learning rate | 10e2 | 10e2 | 10e2 | 10e2 | 10e2 | 10e2 | 10e2 | 10e2 | 10e2 | 10e2 |

Table 3: Hyper-parameters used for the attacks on Adult dataset.

**Additional Dataset Description**

| Airline ID | 0 | 1 | 2 | 3 | 4 | 5 | 6 | 7 | 8 | 9 |
|---|---|---|---|---|---|---|---|---|---|---|
| Local data size | 2995 | 17451 | 11789 | 9855 | 8209 | 8058 | 4232 | 4161 | 3959 | 19287 |
| Dissimilarity degree | 23% | 8% | 4% | 21% | 8% | 4% | 25% | 8% | 5% | 6% |
| Prior $\rho$ | 62% | 66% | 60% | 70% | 63% | 62% | 68% | 55% | 58% | 61% |

Table 4: Statistics of Flight Prices dataset for every airline. Total size: 100 000 records.

**Flight Prices dataset**    To show the dissimilarity between the clients local data distribution, the relative Euclidean distance between each client's local optimum model and the final global model is evaluated [28]. We observe from Table 4 that local data distribution of client 0 and 6 are very different from the global distribution. This corresponds to the fact that client 0 is a business fares airline and client 6 is a basic economy fares airline. The local dataset size for each client and the true prior $\rho$ of the sensitive attribute journey type are given as well in Table 4.

**Adult**    We perform our attacks on a subset of the data where the individual's education level is at least "bachelor".[3] There are 10 clients. To simulate a non-iid data distribution scenario, we distribute the records of people with a PhD degree among the first three clients according to their age. The first client owns the data of young PhDs less than 38 years old, the second client owns the data of PhDs aging between 38 and 52 years old, and the third client owns the data of PhDs elder than 52 years old. The rest of the data is uniformly distributed among the remaining clients. To show the dissimilarity between clients, the relative Euclidean distance between each client's local optimum model is evaluated (See Fig. 2). We can observe that, due to our specific non-iid data distribution, the local models of the first three clients are quite far from the rest of the local models, which is reasonable as people with PhD degree are more likely to have a different salary prediction pattern.

| Client ID | 0 | 1 | 2 | 3 | 4 | 5 | 6 | 7 | 8 | 9 |
|---|---|---|---|---|---|---|---|---|---|---|
| Local data size | 2% | 3% | 2% | 13% | 13% | 13% | 13% | 13% | 14% | 14% |

Table 5: Local data size proportion of each client in Adult dataset. Total size: 12 300 records.

---

[3]The number of the data points are reduced from 48842 to 12300.

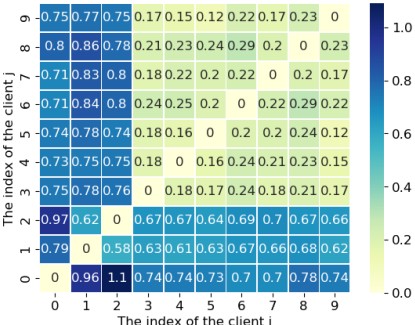

Figure 2: Adult: Clients model heterogeneity, $\frac{\|\theta_i^* - \theta_j^*\|_2}{\|\theta_j^*\|_2}$

**Additional explanations for the experiments**

**The choices for the messages/gradients in attacks**  For the gradient-based attribute inference attack (LBFGS and Gumbel), the choices for the gradients are critical to the attack's performance. We grid-searched as well the above two parameters for the choices of gradients and used the configuration which minimizes the corresponding optimization problem. Remember that LBFGS picks the sensitive attributes that minimize the Euclidean distance between the virtual gradients and the true gradients. Gumbel minimizes the cosine similarity between the virtual gradients and the true gradients. The details on the two parameters after grid-searching for each client are shown in Table 6 for Flight Prices dataset and in Table 7 for Adult dataset.

| | client ID | 0 | 1 | 2 | 3 | 4 | 5 | 6 | 7 | 8 | 9 |
|---|---|---|---|---|---|---|---|---|---|---|---|
| **LBFGS** | Optimal communication rounds | 600 | 500 | 400 | 500 | 800 | 700 | 900 | 900 | 800 | 1000 |
| | Optimal decoding steps | 20 | 20 | 20 | 30 | 30 | 30 | 50 | 40 | 50 | 50 |
| **Gumbel** | client ID | 0 | 1 | 2 | 3 | 4 | 5 | 6 | 7 | 8 | 9 |
| | Optimal communication rounds | 600 | 600 | 400 | 500 | 500 | 700 | 600 | 700 | 800 | 600 |
| | Optimal decoding steps | 30 | 30 | 20 | 30 | 20 | 20 | 50 | 40 | 50 | 30 |

Table 6: Flight prices: Linear regression model - The configuration for the choices of the gradients considered in LBFGS and Gumbel for attribute inference attack. For example, with 300 communication rounds and 10 decoding steps, the adversary chooses the gradients from the communication rounds $\{0, 10, 20, 30, ..., 300\}$.

| | client ID | 0 | 1 | 2 | 3 | 4 | 5 | 6 | 7 | 8 | 9 |
|---|---|---|---|---|---|---|---|---|---|---|---|
| **LBFGS** | Optimal communication rounds | 700 | 600 | 1000 | 800 | 700 | 700 | 900 | 700 | 900 | 1000 |
| | Optimal decoding step | 20 | 20 | 30 | 20 | 40 | 30 | 30 | 10 | 10 | 30 |
| **Gumbel** | client ID | 0 | 1 | 2 | 3 | 4 | 5 | 6 | 7 | 8 | 9 |
| | Optimal communication rounds | 700 | 600 | 900 | 900 | 800 | 700 | 900 | 800 | 900 | 1000 |
| | Optimal decoding step | 20 | 30 | 50 | 40 | 30 | 20 | 40 | 20 | 10 | 30 |

Table 7: Adult: Logistic regression model - The configuration for the choices of the gradients considered in LBFGS and Gumbel for attribute inference attack. For example, with 300 communication rounds and 10 decoding steps, the adversary chooses the gradients from the communication rounds $\{0, 10, 20, 30, ..., 300\}$.

**The dependence of model-based AIA reconstruction performances**  Here we study the effect of the metric $\frac{E}{(\theta^s)^2}$ (where $E$ is the mean sqaure error of the model over the victim's local dataset and $\theta^s$ is the coefficient of sensitive attribute) on our model-based AIA attack accuracy. As shown in Proposition 3, when training linear least square regression task, the smaller this metric is, the higher our model-based attack accuracy is. This theory adapts as well to other model-based attacks which follow the same procedure in Algo. 3 but replace the local model with the final global model, the last returned model or the analytical model.

In Figure 3, we show for each client and each model-based attack, the relation between the AIA accuracy and the corresponding metric $\frac{E}{(\theta^s)^2}$. First, we can see that the attack performance does depend on the metric $\frac{E}{(\theta^s)^2}$, which confirms the lower bound proposed in Prop. 3. For example, client 0 is more vulnerable than the others to the AIA attack (that our attack achieves $97.7\%$ accuracy in Table 1), as the decoded local model has smaller value on the metric $\frac{E}{(\theta^s)^2}$ (blue circle point compared with the other blues). Generally, we observe that for the analytical models (red points) which fit the best the training data enjoy a high reconstruction accuracy (almost all the points are located at the top left). Second, we can observe that the average AIA attack accuracy of our decoded model (blue points) is close to the baseline performance when performing attack on the analytical model (red points). Third, for some airlines (clients 6 and 8 in Table 1), the reason why our decoded models (blue up/down-pointing triangles) perform worse than the global model (green ones) and the last returned models (orange ones), is that our decoded models have a larger $\frac{E}{(\theta^s)^2}$ metric due to their low importance $\theta^s$ on the sensitive feature.

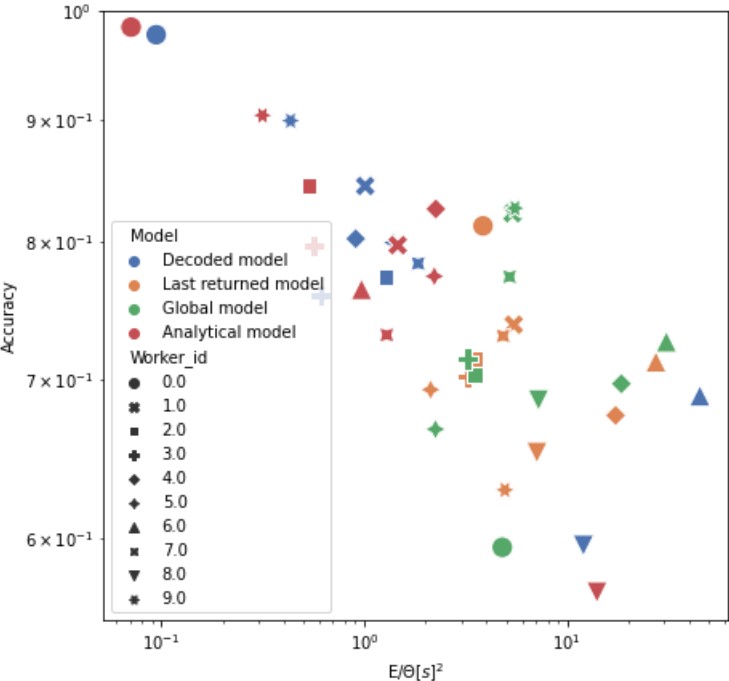

Figure 3: Influence of the metric $\frac{E}{(\theta^s)^2}$ on the AIA accuracy when attacking the global model, the last returned model to the server, the decoded model and the analytical model during the training of linear regression model on Flight Prices dataset through FedAvg with full batch size and one local step.

