# OpenReview forum: "A Novel Model-Based Attribute Inference Attack in Federated Learning"
_NeurIPS.cc/2022/Workshop/Federated_Learning — FL-NeurIPS 2022 Poster_

### Official Review · Reviewer_a1me · 2022-10-08

This paper presents an attribute inference attack in federated learning via exploiting the model communicated between clients and the server instead of the gradients.

Strengths:
- The model based method do not require the small batch size and small number of local training epochs which are assumed in previous gradient-based attacks.
- The theoretical analysis on the failure of gradient-based attacks is very interesting.
- Extensive experiments are conducted.

Weaknesses:
- The theoretical analysis is only performed on very simple model, it is not clear whether the same conclusion can be applied to more complex models (e.g., CNN).
- The evaluation is also only conducted on tabular data and simple models, the results will be more convincing if other data modalities (e.g., images) are considered with more complex models.

---

### Official Review · Reviewer_3BgZ · 2022-10-16
**This paper proposes a model-based attribute inference attack (AIA) for a client’s sensitive attribute in federated learning. It was empirically shown to have good inference accuracy and robustness to increase of data size. The reviewer would suggest reject the paper due to its unrealistic assumptions and limited technical novelty.**

The reviewer has the following major concerns about the paper:
1. As shown in Algorithm 2, in typical FL settings, a client’s local update takes multiple epochs over multiple data batches. So the assumptions of full batch size and one local step will not hold in realistic FL tasks.

2. The technical novelty of the proposed AIA is rather limited, as the major step of the attack is to find the optimal local model using the local model reconstruction attack in [20].

---

### Official Review · Reviewer_zaiw · 2022-10-18
**A Novel Model-Based Attribute Inference Attack in Federated Learning**

This paper proposes a new attribute inference attack on Federated Learning. The proposed attack is model-based which overcomes the limits of gradient-based one.

Strengths:
+ The proposed attack appears to be promising with an improvement of 6 to 10 percentage points in the attack accuracy.

+ The attack assumptions are valid. Paper assumes weak adversary, who is honest-but-curious, i.e, who only eavesdrops the exchanged messages between the client and the server but does not interfere with the training process.

+ Paper presents an interesting insight that there exists a dataset with a binary sensitive attribute on which no gradient-based attribute inference attack in FL can achieve more than 50% accuracy.

+ I like the model-based approach and it appears to be promising.

Weaknesses:
- Evaluation is done on very simple datasets (Flight Prices and Adult datasets). I suggest using bigger and complex datasets.

- Paper claims that the proposed model-based attack performs better under heterogeneous clients. However, important details are missing to back up this claim. Please provide detail explanation how your approach is better for heterogeneous clients compared to existing attacks.

- Table 1 is violating the formatting rules.

---

### Decision · Program_Chairs · 2022-10-20

Accept (Poster)